# Calculation of steel corrosion rate of reinforced concrete slab based on rust expansion crack

**Duo Wu¹, Ziyi Zou¹, Hao Wu¹, Weihong Wan¹, Shangchuan Zhao², Jian Cao◯¹***

**1** School of Civil and Architecture Engineering, Nanchang Institute of Technology, Nanchang, China,
**2** Ministry of Transport Research Institute of Highways, Beijing, China

* caojian1980@126.com

## Abstract

This study aims to develop a refined calculation model that incorporates the effects of distributed reinforcement on crack propagation, validated through experimental and theoretical analysis, to improve the accuracy and applicability of nondestructive corrosion assessments. In this paper, two reinforced concrete slabs were made to consider the influence of distributed reinforcement on the corrosion process of main reinforcement.An accelerated surface cracking method was used, employing reinforcement electrification. This approach tested the relationship between crack width in reinforced concrete slabs and the corrosion rate of main reinforcement. On the basis of the existing calculation model of the relationship between the corrosion rate of reinforcement and the width of concrete surface crack, combined with the development mechanism of concrete surface crack caused by the corrosion expansion of main reinforcement under the lateral constraint conditions, a calculation model of the surface crack width of reinforced concrete slab and the corrosion rate of reinforcement is established. The comparison shows that the calculation results of the model in this paper can better reflect the test rules, and provide a reference for nondestructive quantitative detection of reinforcement corrosion in concrete structures.

Ensuring the durability and safety of reinforced concrete structures has become a critical focus in civil engineering, particularly in the context of increasing environmental challenges and aging infrastructure. With the long-term application of engineering structures, steel reinforcement in concrete is often rusted to varying degrees by the intrusion of chloride ion, making rust expansion and cracking, protective layer peeling and rebar exposing and so on [1]. The loss of reinforcement cross-section in rusted members will degrade not only the mechanical properties of reinforcement but also the bond between reinforcement and concrete, besides it lead to the appearance and expansion of cracks on the surface of the structure, which seriously endangers the safety, serviceability and durability of structures [2]. Therefore, it is of practical

**Data availability statement:** All relevant data are within the paper and its Supporting Information files.

**Funding:** This research was supported by the National Natural Science Foundation of China (Grant No.52168030) awarded to J.C. and the Natural Science Foundation of Jiangxi Province (Grant No.20232BAB204068) awarded to J.C.

**Competing interests:** The authors have declared that no competing interests exist.

engineering significance to quantitatively grasp the extent of reinforcement corrosion within concrete for accurate assessment of structural reliability. Corrosion in reinforced concrete structures is a critical challenge requiring precise detection and analysis. Recent advancements in corrosion monitoring techniques, as reviewed by Meyer et al. [3], emphasize the growing application of non-destructive methods such as acoustic emission and electrochemical techniques for real-time corrosion assessment. This study builds on these advancements by integrating the role of distributed reinforcement into the analysis of crack behavior.

Three types of methods are commonly used in engineering to detect the amount of reinforcement corrosion, including nondestructive testing methods, semi-(micro) breakage testing methods and comprehensive testing methods [4]. Nondestructive testing methods are testing techniques based on the correlation between the amount of reinforcing steel corrosion and some physical quantities without affecting any performance of the structure or member, and can be broadly classified into physical, acoustic emission and electrochemical testing methods. Among them, acoustic emission and electrochemical methods are faster, but the test accuracy can be affected by various factors such as internal defects in concrete, multiple and complex signal sources, and uneven distribution of rust reactants [5]. Therefore, based on the methods of concrete material mechanics and elastic mechanics analysis, the evolution of crack width on the surface of the structure is used to characterize the amount of corrosion process of reinforcement, which can meet the accuracy needs of practical engineering nondestructive testing.

From the point of view of satisfying the regulations of the limit state of normal use of the member, there exists a limit to the width of cracks on the surface of the member, but before this limit is reached, it is possible that the safety or durability of the structure has been destroyed. At present, researchers have carried out a lot of experimental and theoretical studies around the corrosion process of reinforcement in concrete, and have given the evolution law and calculation models of the amount of corrosion of reinforcement at different stages [6–11]. Cao et al. [12] explored the interaction between crack expansion and corrosion medium propagation caused by reinforcement corrosion and established a theoretical model of reinforcement corrosion amount in the coupled microscopic and macroscopic chloride corrosion process. Papakonstantinou et al. [13] considered the actual size of the structure in engineering and the environment in which it is located, and based on the size effect and reliability theory, proposed a large size Li et al. [14–17] carried out a series of theoretical and experimental studies on the surface cracks of reinforced concrete members caused by rust expansion, using the theory of concrete material mechanics and elasticity mechanics, and gave a theoretical model of the relationship between the crack width and the amount of corrosion of reinforcing steel considering the softening of concrete materials after cracking. However, the shortcomings are that the expansion behavior of concrete rust swelling cracks is very complicated and the understanding of concrete rust swelling mechanism is not comprehensive enough, which leads to the complicated calculation process of the existing models, the existence of large dispersion in the calculation results, and the difficulty in determining the model parameters. Cao et al.

[18] calculated the stiffness discount factor of concrete in the model based on the available experimental data, considered the reinforcement diameter, concrete protective layer thickness and water-cement ratio and other parameters, an engineering equation for the degree of reinforcement corrosion related to the width of cracks on the surface of reinforced concrete was established. The results of the studies have shown that the degree of corrosion of the reinforcement has a significant effect on the width of surface cracks of the members and is closely related to the parameters such as the strength grade of the concrete, the size of the members, the thickness of the concrete protective layer and the diameter of the reinforcement. However, the differences in test methods and test conditions lead to the lack of comparability and regularity of test results.

In view of the fact that most of the current studies on the relationship between the amount of reinforcement corrosion and the width of cracks on the surface of the structure only consider the case where only the main reinforcement exists within the concrete, while ignoring the effect of the actual reinforcement allocation. Yang et al. [19] found through experiments that there is an obvious linear relationship between the total crack width and the amount of longitudinal reinforcement corrosion for rust swelling cracks along the longitudinal reinforcement direction only; for bidirectional rust swelling cracks along the longitudinal reinforcement and hoop direction, there is no obvious linear relationship between the total crack width and the amount of longitudinal reinforcement corrosion. Jin et al. [20] established a three-dimensional mesoscale model of concrete to study the rust cracking behavior of the concrete protective layer. Zhao et al. [21] considered the influence of distribution reinforcement and stress state, and derived the formula for the amount of reinforcement corrosion in reinforced concrete beams The formula for calculating the amount of corrosion in reinforced concrete beams was derived and verified through tests. The current studies considering the reinforcement condition all consider the effect of the hoop factor within the reinforced concrete beam, while the study of the effect of distributed reinforcement in reinforced concrete slabs has not been reported yet.

Since the width of rust crack is easy to obtain in actual structural inspectionit is also one of the basic parameters reflecting whether the suitability of the structure is satisfied or not. Therefore, the nondestructive testing method of measuring the crack width on the surface of a member without damaging the structural safety is of great significance to track the expansion behavior of rust crack and to grasp the dynamic law of corrosion of reinforcement in concrete. While significant progress has been made in understanding the relationship between corrosion and crack propagation in concrete structures, existing models [12–17] often neglect the influence of distributed reinforcement, which plays a crucial role in practical scenarios. This gap limits the applicability of these models in realworld conditions.

In this paper, firstly, the experimental study on the corrosion amount of main reinforcement in reinforced concrete slab with different rust crack widths is carried out considering the parameters of whether to arrange distributed reinforcement and protective layer thickness, and the experimental phenomena and results are analyzed and discussed; secondly, based on the existing study on the relationship between the corrosion amount of reinforcement and the surface crack width of the member, the calculation analysis of the influence of the configuration of transversely restrained hoop on the corrosion of main reinforcement is combined with the calculation analysis of the influence of distributed reinforcement to establish the relationship between the corrosion amount of reinforcement and the surface crack width of the member. Finally, the model is compared with the experimental data in this paper, and suggestions are made on the applicability of the model.

The key contributions of this study include: 1) developing a novel calculation model that accounts for distributed reinforcement, 2) conducting experimental validation to ensure the model's practical reliability, and 3) providing insights for non-destructive testing methods applicable to real-world engineering.

## 1 Modeling

### 1.1 Model without distributed reinforcement

As mentioned above, numerous researchers have proposed calculation models [12–17] that consider the relationship between the corrosion rate of steel bars and the width of surface cracks in components. The author also proposed corresponding calculation models [18], which have been validated. Based on the material mechanics and elasticity theory of

concrete, Li et al. [14] analyzed the expansion cracking region of the damaged protective layer of concrete resulted from steel corrosion, considering concrete with embedded rebar to be a thick wall cylinder, as shown in Fig 1(a–c).

In Li-CQ's model [14], the calculation process of parameters $d_s(t)$ and $\alpha$ is an iterative process, which is more complicated and inconvenient. In addition, when difference method is used to calculate the stiffness reduction factor in the Li-CQ's model [14], the accuracy of calculation results could not be guaranteed if the difference step is large, while the computation burden will be doubled if the difference step is small. Therefore, parameters in the Li-CQ's model should be simplified for engineering applications.

Cao et al. [18] analyzed the area affected by rust swelling cracking damage of the concrete protective layer, and established corrected model of steel corrosion degree related to the surface crack width of the member by introducing a concrete stiffness discount factor.

$$W_{rust}^{\alpha}(t) = \frac{\left(w_c + \frac{2\pi b f_t}{E_{ef}}\right) \cdot (D + 2d_0) \cdot (1 - \nu_c)(a/b)^{\sqrt{\alpha'(t)}}}{4 \cdot \left(\frac{1}{\rho_{rust}} - \frac{\alpha_{rust}}{\rho_{st}}\right)} + \frac{\left(w_c + \frac{2\pi b f_t}{E_{ef}}\right) \cdot (D + 2d_0) \cdot (1 + \nu_c)(b/a)^{\sqrt{\alpha'(t)}}}{4 \cdot \left(\frac{1}{\rho_{rust}} - \frac{\alpha_{rust}}{\rho_{st}}\right)}$$

(1)

Where $E_{ef}$ is Effective elastic modulus; $D$ is Rebar diameter; $d_0$ is thickness of the annular zone of concrete pores between the interface of reinforcement and concrete bond; $\nu_c$ is the Poisson's ratio of concrete; $C$ is the thickness of concrete protection layer; $a = (D + 2d_0)/2$; $b = C + (D + 2d_0)/2$; $\alpha'(t)$ denotes modified concrete stiffness reduction factor; $\rho_{rust}$ is the density of rust products; $\rho_{st}$ is rebar density; $\alpha_{rust}$ is the coefficient related to the type of rust product. The parameters are specifically calculated in the literature [18].

From the formula (1), it can be seen that the width of the crack on the surface of the component is affected by the parameters of the concrete material properties, the diameter of the reinforcement as well as the thickness of the concrete protective layer, besides, the calculation formula is based on the concrete material mechanics and elasticity theory, which possesses a clear physical meaning and high calculation accuracy, therefore. This paper bases on formula (1), considering the effect of distributed reinforcement on the crack development, and establishes the corresponding calculation model.

For modified concrete stiffness reduction factor $\alpha'(t)$, considering the influence of the corrosion time of steel bar, the data in the literature [15,22–34] were applied to calculate the simplified calculation of $\alpha'(t)$, and the nonlinear regression Equation can be indicated as follows

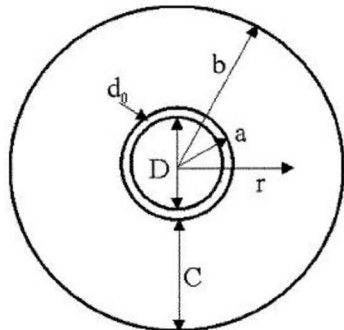

(a) reinforced concrete considered to be a thick wall cylinder

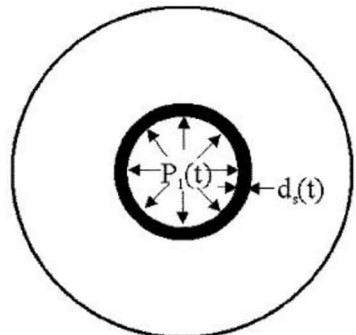

(b) expansion stress caused by corrosion products

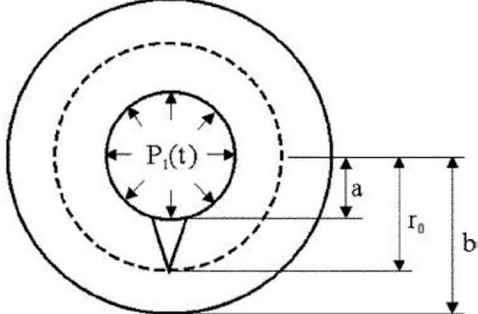

(c) concrete cover cracking

**Fig 1. Schematic of crack evolution.**

$$\alpha'(t) = 0.0114 \times (t/365)^{(-2.199)} + 0.124 \qquad \left(adj.\ R^2 = 0.97\right) \tag{2}$$

Where, $t$ is the time of steel corrosion, day as the unit.

From Equation (1) and (2), it can be seen that the stiffness degradation coefficient of concrete increases with the increase of corrosion time, and the accumulation of corrosion products is directly proportional to the width of surface cracks in the component.

Consequently, substituting the Equation (2) into the Equation (1), the modified parameter of the mass of corrosion products $W^\alpha_{rust}(t)$ can be proposed as follows

$$W^\alpha_{rust}(t) = \frac{\left(w_c + \frac{2\pi bf_t}{E_{ef}}\right) \cdot (D + 2d_0) \cdot \left[(1-\nu_c)(a/b)^{\sqrt{\alpha'(t)}} + (1+\nu_c)(b/a)^{\sqrt{\alpha'(t)}}\right]}{4 \cdot \left(\frac{1}{\rho_{rust}} - \frac{\alpha_{rust}}{\rho_{st}}\right)} \tag{3}$$

Thus, the modified calculation of steel corrosion degree can be expressed as

$$\theta^\alpha = \frac{W^\alpha_{rust}}{W} \times 100\% \tag{4}$$

Where, $\theta^a$ is the degree of steel corrosion (%); $W$ is is the initial mass of steel bar.

## 1.2 Effect of distributed reinforcement

In actual structural components, such as reinforced concrete slabs, the stressed steel bars are located inside the distributed steel bars. Before the steel bars under stress corrode, they are exposed to external environmental corrosion earlier due to their closer proximity to the structural surface. Therefore, when analyzing the corrosion law of stressed steel bars, it is also necessary to consider the influence of early corrosion of distributed steel bars.

When the distributed steel bars undergo corrosion earlier, the corrosion products on the surface of the distributed steel bars will cause concrete cracking, and tensile stress will also be generated at the contact area with the stressed steel bars, resulting in a significant difference in the development process of corrosion of the stressed steel bars compared to

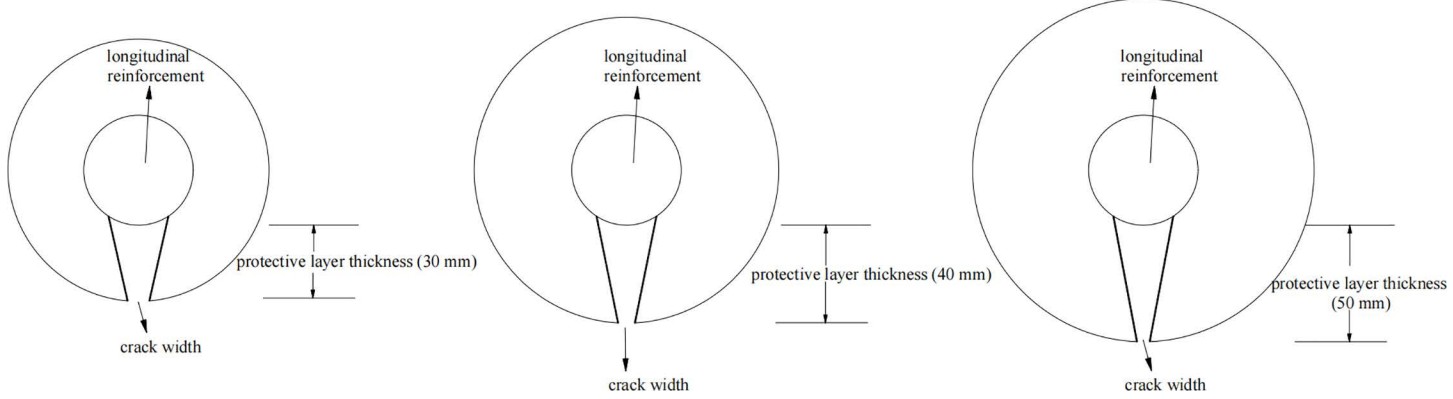

**Fig 2. The influence of different protective layer thicknesses on crack development.**

the absence of distributed steel bars. The mechanism of crack development in concrete under different thicknesses of protective layers is shown in Fig 2.

Chen et al. [35] considering the influence of distribution reinforcement on the expansion of concrete rust swelling cracks, gave the formula for calculating the width of concrete surface cracks with and without distribution reinforcement, which based on the deformation coordination conditions before and after the rust swelling cracking of concrete in the protective layer.

When the hoop is located at the corner:

$$\frac{w_{with}}{w_{without}} = \frac{l_{sv}}{(EA)_{sv}} \cdot \frac{1}{\frac{l_{sv}}{(EA)_{sv}} + \frac{d^3}{8E_cI}\left(\frac{9\pi}{4} + 2\right) + \frac{3\pi d}{8E_cA} + \frac{3\pi kd}{8GA} + \frac{\pi d}{8(E_cA + E_{sv}A_{sv1})}} \tag{5}$$

When the hoop is in the middle:

$$\frac{w_{with}}{w_{without}} = \frac{l_{sv}}{(EA)_{sv}} \cdot \frac{1}{\frac{l_{sv}}{(EA)_{sv}} + \frac{3\pi d^3}{8E_cI} + \frac{\pi d}{2E_cA} + \frac{\pi kd}{2GA}} \tag{6}$$

In formula (5) and (6)$G$ is the shear modulus of concrete, $G = \frac{E_c}{2(1+\mu)} = \frac{E_c}{2(1+0.2)} = 0.42E_c$; A indicates the area of concrete affected by distribution reinforcement, $A = c \cdot s$, where $c$ is the thickness of concrete protection layer, $s$ indicates hoop spacing; $I = (1/12)s \cdot c^3$; $k = 1.2$ (taking rectangular section); $d$ is hoop diameter; $A_{sv1}$ is cross-sectional area of single-limb distribution reinforcement, $E_{sv}$ is the modulus of elasticity of distribution reinforcement. Let $\lambda = \frac{d}{c}$, $\beta = \frac{E_{sv}}{E_c}$, $\rho'_{sv} = \frac{A_{sv}}{b \cdot s}$, and then:

When the hoop is located at the corner:

$$\frac{w_{with}}{w_{without}} = \frac{1}{1 + \beta\frac{13.608\lambda^3 + 4.544\lambda + 0.393\frac{1}{\lambda + \beta\rho'_{sv}b/d}}{\frac{l_{sv}}{\rho'_{sv}b}}} \tag{7}$$

When the hoop is in the middle:

$$\frac{w_{with}}{w_{without}} = \frac{1}{1 + \beta\frac{14.136\lambda^3 + 6.059\lambda}{\frac{l_{sv}}{\rho'_{sv}b}}} \tag{8}$$

Thus, combining formula (1) and (8), the calculation model of rust expansion crack width and reinforcement corrosion amount considering the influence of distributed reinforcement can be obtained. The calculation process of modified model includes the following steps.

1) Determine the basic parameters of reinforced concrete members, the relationship between the steel corrosion product thickness $d_s(t)$ and concrete stiffness reduction factor $\alpha$ can be calculated;

2) Substitute the relationship with $d_s(t)$ and $\alpha$ to the Equation (2), the relationship between the crack width $w_c$ and $d_s(t)$ can be obtained.

3) Substitute $d_s(t)$ represented by $w_c$ to the Equation (1), the relationship between $w_c$ and $W_{rust}(t)$ can be established. Further, the model of the corrosion degree of steel bar can be calculated by Equation (3);

4) Finally, by considering the reinforcement parameters of the slab and combining the Equation (1), the Equation (7) and the Equation (8), the calculation model of rust expansion crack width and reinforcement corrosion amount considering the influence of distributed reinforcement can be obtained.

Building on the work of Chen et al. [34], this study advances the understanding of distributed reinforcement by introducing an adjusted stiffness reduction factor and experimental validation. For instance, the presented Equation (3) refines the crack-width prediction, accommodating nonlinear effects observed during accelerated corrosion tests. Moreover, it is explicitly state that the model is intended to link crack width with reinforcement corrosion and assess its predictive power compared to experimental data.

Unlike previous models [12–17], this study combines distributed reinforcement and its early corrosion effects, providing a more comprehensive framework for evaluating crack propagation and reinforcement corrosion in reinforced concrete. This allows engineers to predict crack width and corrosion rate, which helps to proactively maintain and extend the life of critical infrastructure.

It is worth pointing out that although this article considers the influence of distributed steel bars in actual components on the corrosion process of stressed steel bars, there are still certain shortcomings, such as neglecting the effects of external stresses or environmental variability, and using a simplified corrosion process in calculations.

## 2 Experiment

The experimental setup was designed to investigate the effects of protective layer thickness and distributed reinforcement spacing on crack width and reinforcement corrosion. Using a semi-wet electrification method, the study aimed to replicate accelerated corrosion conditions while maintaining relevance to practical engineering scenarios.

The primary objective of the experimental setup is to validate the proposed model by analyzing the relationship between surface crack width and reinforcement corrosion under different protective layer thicknesses and distributed reinforcement configurations.

### 2.1 Reinforced concrete slab specimens

In this test, two reinforced concrete slabs were fabricated. I# slab shown in Fig 3 is 1 000 mm (length) × 500 mm (width) × 200 mm (thickness), of which the reinforcement protective layer thickness is 30 mm, 40 mm and 50 mm respectively, and

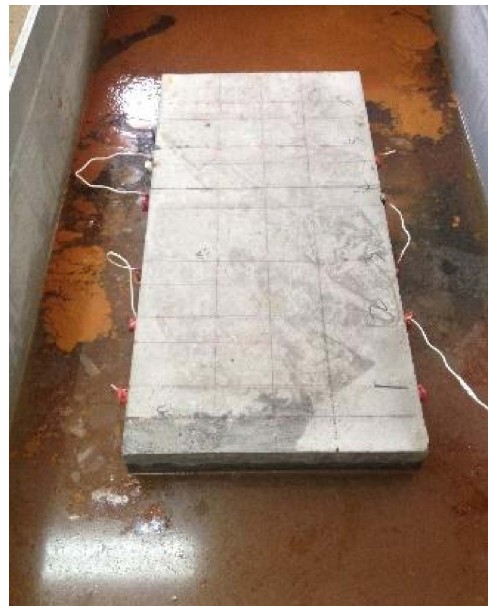

**Fig 3. I# Reinforced concrete slab.**

the diameter of steel bars is 25 mm, the long of each steel bar is 560 mm, without distributed reinforcement in the I# slab. The concrete strength of I# slab is C50. Details is shown in the following schematics, as shown in the Figs 4–6.

II# slab shown in Fig 7 is 1 000 mm (length) × 700 mm (width) × 200 mm (thickness), of which the protective layer thickness of reinforcement is 40 mm. The diameter of steel bars is 25mm, the long of each steel bar is 1080 mm, with the diameter of distributed reinforcement is 10 mm in the II# slab. The spacing between each distributed steel bar are 100 mm

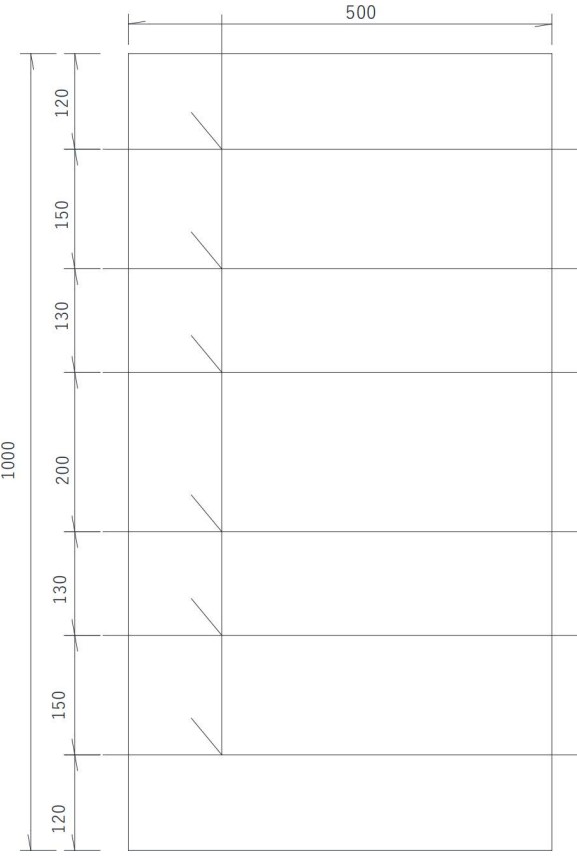

**Fig 4. Top view of I# slab (unit:mm).**

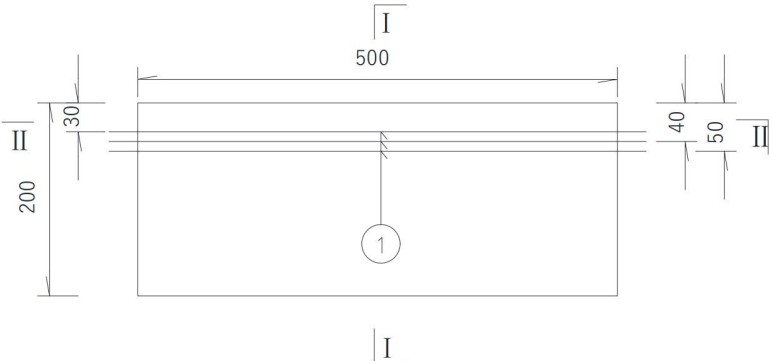

**Fig 5. Front elevation view of I# slab (unit:mm).**

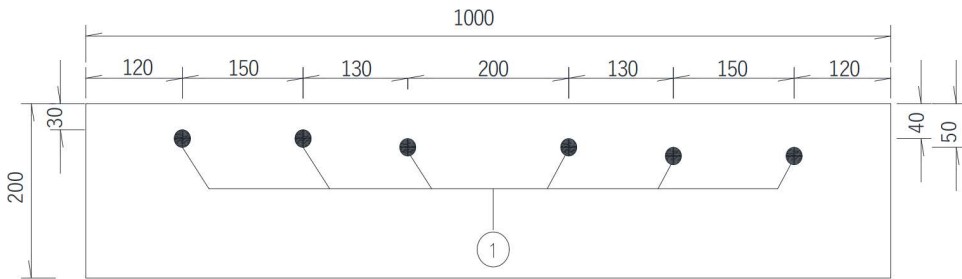

**Fig 6. Side elevation view of I# slab (unit:mm).**

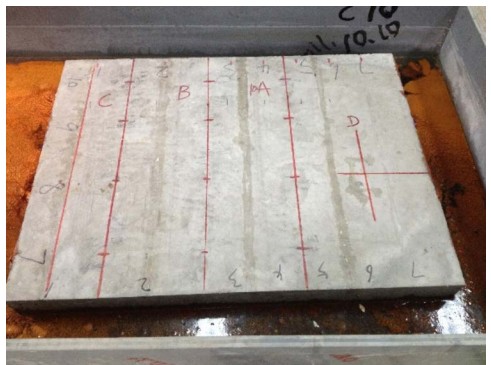

**Fig 7. II# Reinforced concrete slab.**

and 200 mm respectively in the II# slab. The concrete strength grade of II# slabs is also C50. Details are shown in the following schematics, as shown in the Figs 8–10.

In the Figs 7–10, ①, ②, ③ represent the diameter of steel bars is 25 mm in I# slab, the diameter of steel bars is 25 mm in II# slab and the diameter of steel bars is 10 mm in II# slab respectively.

## 2.2 Design of energized rust

In this test, the accelerated rusting of reinforcing steel in concrete slab specimens was carried out by the semi-wet electrification method. First, the reinforced concrete slab after 28d of curing was removed from the curing room with surface cleaned. Then, the exposed main reinforcement at the outer end of the slab was ground smooth to facilitate the welding of the conductors. After the welding of the conductor is completed, the area near the exposed reinforcement and the main reinforcement at both ends of the slab is sealed with epoxy resin. After the epoxy resin dries, the reinforced concrete slab is placed into a steel water tank with a 5% mass fraction of NaCl solution, and the liquid level of the solution is always kept at about half of the slab thickness (about 10 cm), during the test. Meanwhile, the outer wall of the water tank is connected to the negative terminal of the DC power supply, and the wire connected to the main reinforcement is connected to the positive terminal of the DC power supply. At the same time, the salt solution was replenished daily to ensure that the reinforced concrete slab was in a semi-wet corrosive environment, and the accelerated corrosion test was conducted for a total of 7 days. The energized corrosion test is shown in Fig 11.

Figure 11 illustrates the accelerated corrosion setup, which ensures uniform NaCl distribution and maintains consistent semi-wet conditions. Observations from this setup, such as bubble formation, confirm the ongoing corrosion process.

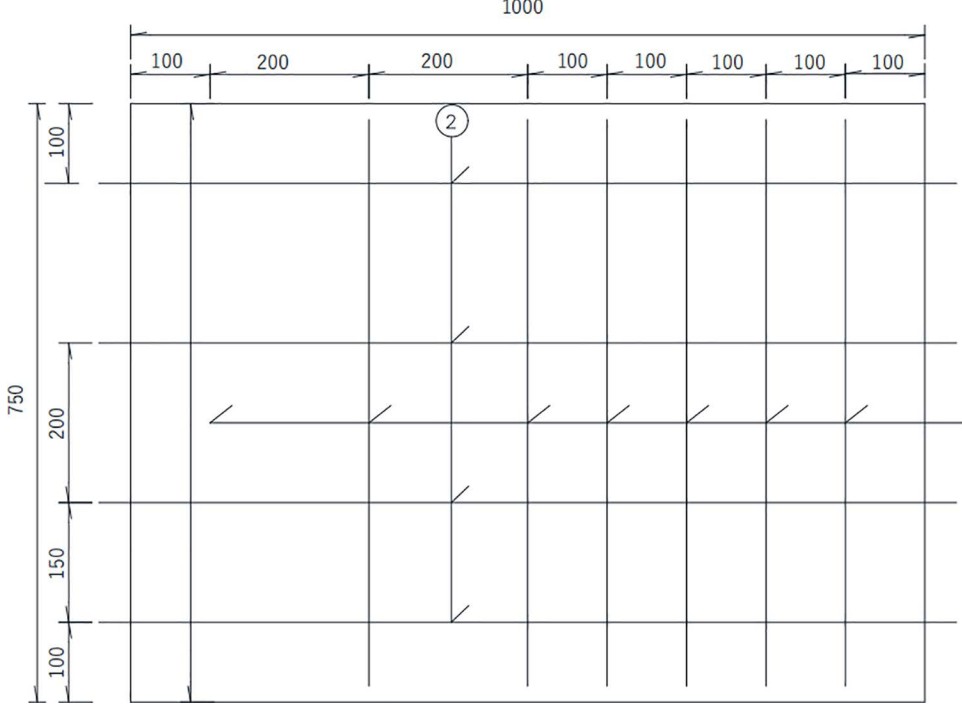

**Fig 8. Top view of II# slab (unit:mm).**

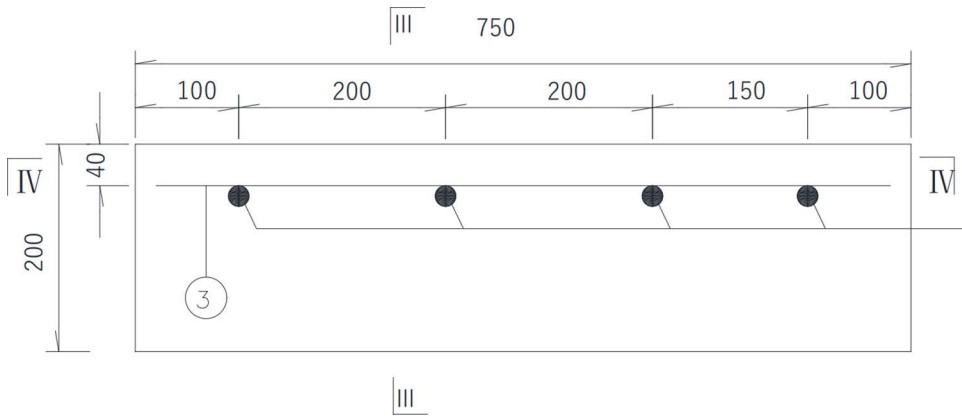

**Fig 9. Front elevation view of II# slab (unit:mm).**

## 3 Results of the experiment

The experimental results reveal critical insights into the effects of protective layer thickness and distributed reinforcement on reinforcement corrosion and surface crack width. Tables 1 and 2 summarize the test data, while Table 3, Figs 12 and 13 compare experimental values with theoretical predictions.

After connecting the reinforcing steel inside the two reinforced concrete slabs to the power supply, a large number of bubbles could be observed appearing around the reinforced concrete slabs, indicating that the circuit had been connected and the reinforcing steel started to corrode rapidly. During the test, the test is checked for normal every 12 hours,

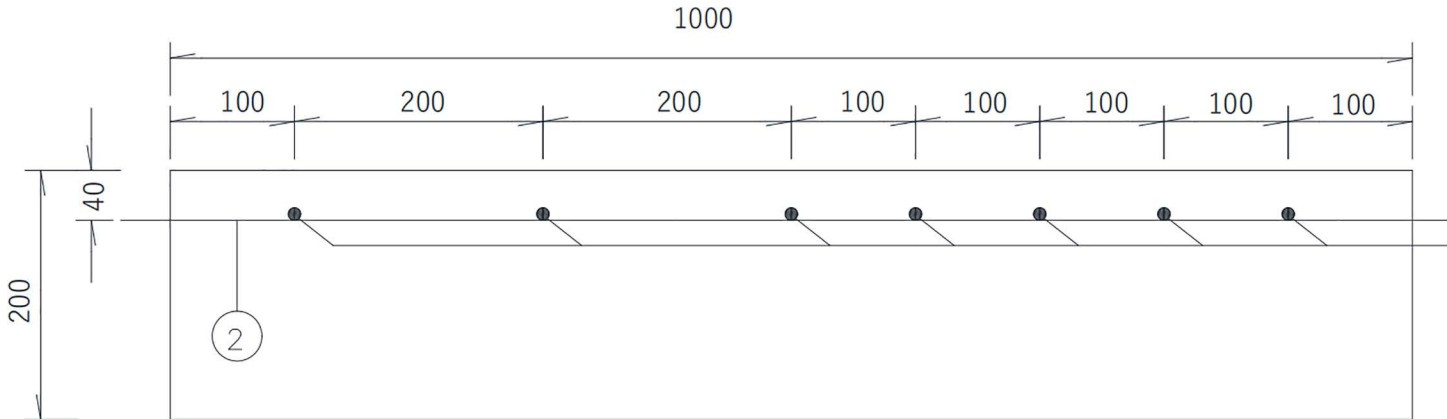

**Fig 10. Side elevation view of Ⅱ# slab (unit:mm).**

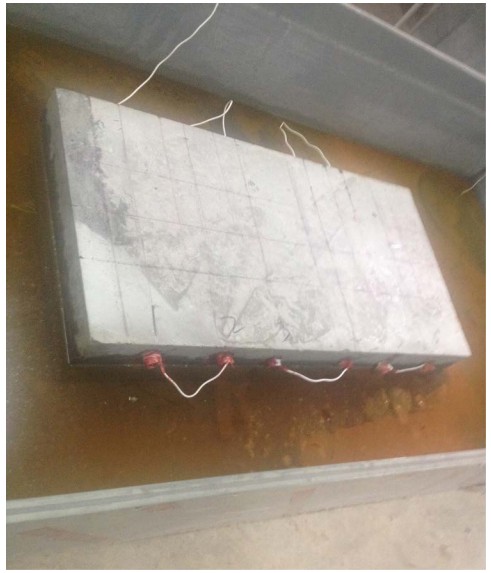

**Fig 11. Reinforced concrete slab energized corrosion test.**

**Table 1. Effects of protective layer thickness (I# slab).**

| Rebar number | Rebar length/(mm) | Rebar diameter/(mm) | Uncorroded weight/(kg) | Corroded weight/(kg) | Corresponding protective layer thickness/(mm) | Amount of rebar corrosion/(%) | | Width of cracks/(mm) | |
|---|---|---|---|---|---|---|---|---|---|
| 1 | 500 | 25 | 1.716 | 1.688 | **30** | 1.63 | **1.63 (avg)** | 0.21 | **0.205 (avg)** |
| 2 | 500 | 25 | 1.721 | 1.693 | | 1.63 | | 0.20 | |
| 3 | 500 | 25 | 1.721 | 1.692 | **40** | 1.66 | **1.46 (avg)** | 0.24 | **0.185 (avg)** |
| 4 | 500 | 25 | 1.716 | 1.695 | | 1.26 | | 0.13 | |
| 5 | 500 | 25 | 1.712 | 1.698 | **50** | 0.79 | **1.225 (avg)** | 0.11 | **0.155 (avg)** |
| 6 | 500 | 25 | 1.719 | 1.691 | | 1.66 | | 0.20 | |

Table 2. Influence of spacing between distributed reinforcement (II# slab).

| Rebar number | Rebar length/(mm) | Rebar diameter/(mm) | Uncor-roded weight/(kg) | Corroded weight/(kg) | Corresponding protective layer thickness/(mm) | Diameter of distributed rebar/(mm) | Spacing of distributed rebar/(mm) | Amount of rebar corrosion/(%) | | Width of cracks/(mm) | |
|---|---|---|---|---|---|---|---|---|---|---|---|
| 7 | 1000 | 25 | 3.433 | 3.415 | 40 | 10 | 100 | 0.53 | **0.54 (avg)** | 0.11 | **0.115 (avg)** |
| 8 | 1000 | 25 | 3.434 | 3.416 | 40 | 10 | | 0.55 | | 0.12 | |
| 9 | 1000 | 25 | 3.433 | 3.412 | 40 | 10 | 200 | 0.6 | **0.65 (avg)** | 0.15 | **0.165 (avg)** |
| 10 | 1000 | 25 | 3.44 | 3.461 | 40 | 10 | | 0.7 | | 0.18 | |

Table 3. Analysis of the model calculated values and test data values of I# slab and II# slab.

| I# slab | | | | II# slab | | | |
|---|---|---|---|---|---|---|---|
| Rebar number | Measured value (%) | Calculated value (%) | Deviation (%) | Rebar number | Measured value (%) | Calculated value (%) | Deviation (%) |
| 1 | 1.63 | 1.52 | 6.75 | 7 | 0.53 | 0.48 | 9.43 |
| 2 | 1.63 | 1.5 | 7.98 | 8 | 0.55 | 0.49 | 10.91 |
| 3 | 1.66 | 1.56 | 6.02 | | | | |
| 4 | 1.26 | 1.11 | 11.9 | 9 | 0.6 | 0.53 | 11.67 |
| 5 | 0.79 | 0.72 | 8.86 | 10 | 0.7 | 0.64 | 8.57 |
| 6 | 1.66 | 1.54 | 7.23 | | | | |

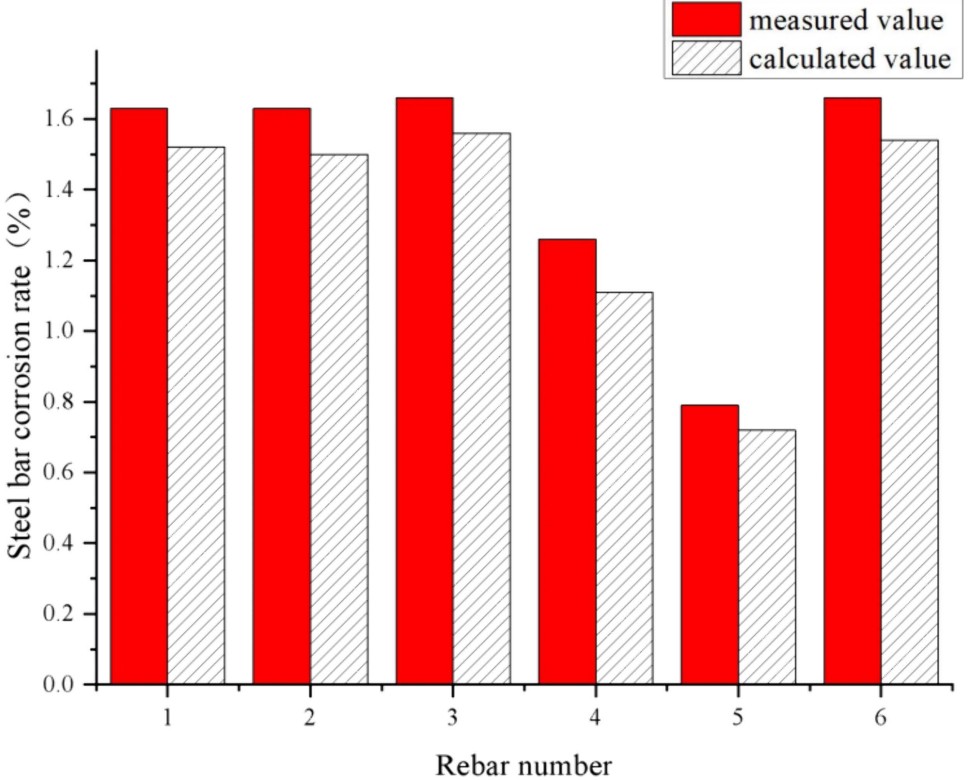

Fig 12. Comparison of test and calculated values of reinforcement corrosion rate in I# slab.

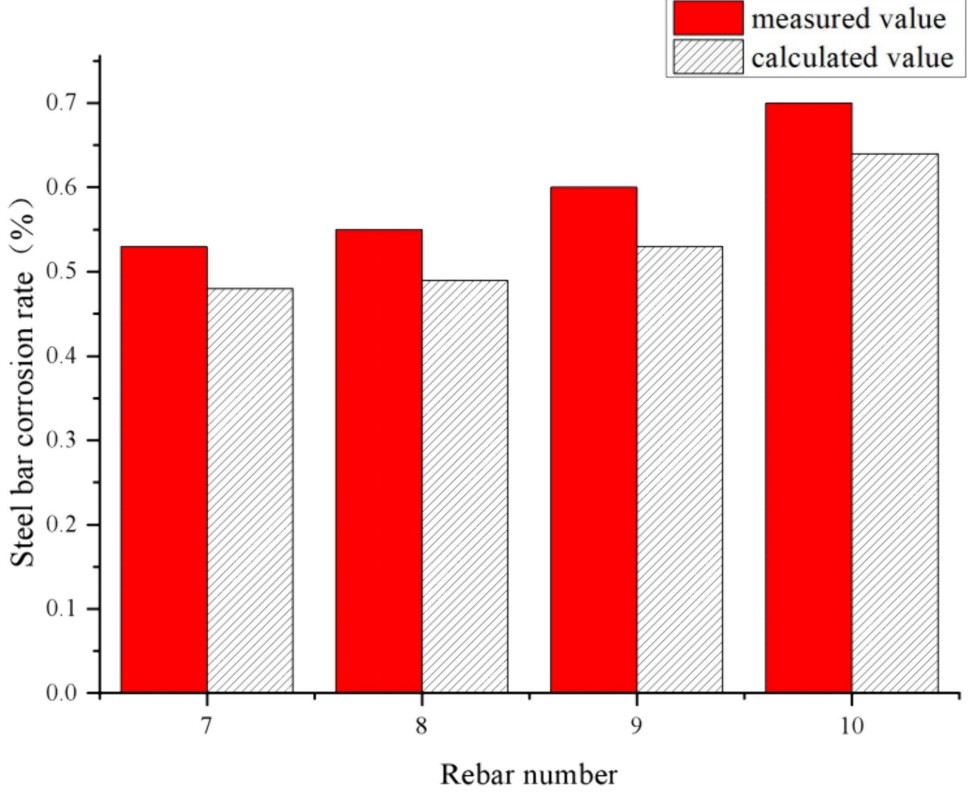

**Fig 13. Comparison of test and calculated values of reinforcement corrosion rate in II# slab.**

including: (1)uniformly stirring the NaCl solution in the steel water tank to ensure uniform distribution of salt solution concentration; (2) always keeping the solution level height in the steel water tank at the set value; (3) observing the bubble situation around the plate and checking whether the current is stable with a multi meter; (4) Clean up the rust products attached to the exposed reinforcement on both sides of the plate every day to ensure uniform rust rate. After 7 days, a thin straight line in the direction of the reinforcement was clearly observed before the crack of rust expansion of the test slab appeared, after which cracks appeared on the surface of the concrete slab. I# slab and II# slab concrete surface crack width, the amount of corrosion of reinforcement and other test data are shown in Tables 1 and 2.

The results demonstrate a clear relationship between protective layer thickness, distributed reinforcement spacing, and the extent of reinforcement corrosion and surface crack width.

From Table 1, for I# slab without distributed reinforcement and with a main reinforcement diameter of 25 mm, the amount of main reinforcement corrosion decreases with the increase in the thickness of the protective layer, namely the average values of the corresponding main reinforcement corrosion were 1.63%, 1.46% and 1.225% when the thickness of the protective layer were 30 mm, 40 mm and 50 mm, respectively, which indicates the concrete protective layer can effectively retard the process of chloride ion invasion in salt solution. As shown in Table 1, the corrosion rate decreases as the protective layer thickness increases, reducing crack width by an average of 25% between 30 mm and 50 mm layers. In addition, when the average values of main reinforcement corrosion were 1.63%, 1.46% and 1.225%, the average values of crack width were 0.205 mm, 0.185 mm and 0.155 mm, respectively, which indicated the crack width on the slab surface decreased with the decreasing of reinforcement corrosion.

In Table 2, for II# slab configured with distributed reinforcement, when the concrete protective layer thickness was 40 mm, the amount of reinforcement corrosion increases with the increase of concrete surface crack width for the same

reinforcement diameter. Taking the average value of the test as the subject of discussion, the concrete surface crack widths were 0.115 mm and 0.165 mm when the distributed reinforcement spacing were 100 mm and 200 mm as well as the corresponding reinforcement corrosion amounts were 0.54% and 0.65%, respectively.

Table 1 highlights the correlation between protective layer thickness and reinforcement corrosion, indicating that thicker layers effectively reduce the corrosion rate and crack width. The experimental findings in Table 2 align with the predictions of Equation (3), particularly in cases where distributed reinforcement spacing was 100 mm.

Besides, comparing with I# slab with the same protective layer thickness without distributed reinforcement, when the spacing of distributed reinforcement were 200 mm and 100 mm, the amount of reinforcement corrosion decreased by 55.5% and 63.0% as well as the width of cracks on the surface of the plate decreased by 10.8% and 37.8%, which indicated that the configuration of distributed reinforcement could effectively reduce the amount of main reinforcement corrosion and reduce the process of cracks on the surface of the plate caused by reinforcement corrosion. The crack development process on the surface of the slab caused by the corrosion of reinforcement could be reduced.

The presence of distributed reinforcement in the II# slab reduced the overall corrosion rate by 55.5% compared to the I# slab, as shown in Table 2. This demonstrates the effectiveness of distributed reinforcement in mitigating crack development. The results in Table 2 indicate that reducing the spacing of distributed reinforcement from 200 mm to 100 mm decreases the corrosion rate by 17%, suggesting that closer reinforcement spacing improves structural durability.

These findings suggest that increasing protective layer thickness and optimizing distributed reinforcement spacing can significantly enhance the durability of reinforced concrete structures exposed to chloride environments. It should be noted that variations in corrosion rates between samples may be attributed to inconsistencies in the curing process or differences in the distribution of the NaCl solution during testing.

Although accelerated corrosion methods may not fully replicate the long-term effects of natural environmental conditions on concrete structures, limiting the generalizability of research results, these experimental insights provide a framework for engineers to design reinforced concrete structures with optimized crack control, thereby extending their service life.

## 4 Modeling verification

This section validates the accuracy and applicability of the proposed model by comparing theoretical predictions with experimental data and practical engineering results.

The proposed model was verified through two approaches: comparison with experimental data and application to a real-world engineering case. The experimental results confirmed the model's accuracy within an acceptable deviation range, while the practical application demonstrated its relevance to engineering practice.

### 4.1 Experimental verification

Using the above model to calculate the amount of corrosion for different test bar numbers respectively, the comparison with the test values are shown in Table 3, Figs 12 and 13.

From Table 3, Figs 12 and 13, for the reinforced concrete slab without and with distributed reinforcement, the calculated values of the theoretical equations constructed in this paper are slightly smaller than the measured values. For the case without distributed reinforcement, the minimum deviation of both is 6.02% and the maximum deviation is 11.90%, with deviations range from 6.0% to 12.0%; for the case with distributed reinforcement, the minimum deviation is 8.57% and the maximum deviation is 11.67%, with deviations range from 8.0% to 12.0%, it is indicated that the accuracy of the proposed model is relatively accurate which can effectively predict the amount of reinforcement corrosion. Considering the calculated value is smaller than the measured value in both cases, it is suggested to add a correction factor to the calculation model.

As shown in Table 3, Fig 12 and 13, the proposed model predicts corrosion rates with deviations ranging from 6% to 12% for slabs without distributed reinforcement and 8% to 12% for slabs with distributed reinforcement, demonstrating its reliability and accuracy.

Compared with widely used models [12–17], the model established in this paper that considers the influence of distributed steel bars not only reduces computational difficulty, but also has better alignment with experimental results under practical conditions. It should be pointed out that the calculation model obtained in this study can better reflect the corrosion mechanism of steel bars in actual engineering components.

### 4.2 Practical engineering verification

It should be noted that in practical engineering, cracks caused by the corrosion of stressed steel bars are generally parallel to the direction of concrete stress caused by external loads on the structure. That is, the influence of concrete stress caused by external loads on the width of corroded cracks is mainly due to the transverse deformation perpendicular to the crack direction. Due to the Poisson's ratio of concrete being approximately between 0.2 and 0.3, and its very small deformation under load, with ultimate strain ranging from 0.002 to 0.003, and working stress generally less than 0.4 times the ultimate stress, its lateral deformation has a very small impact on the macroscopic crack width visible to the naked eye. Due to its minimal impact, the influence of stress state factors on the theoretical model established in this article is not currently considered.

The Sanhongqi Overpass in Shunde City, Guangdong Province, China was built in December 1995. The overall of the bridge is as shown in Fig 14. The bottom part of the bridge slab was designed with A16 load-bearing steel bars, and the diameter of the distributed steel bars is 14 mm, with a spacing of 200 mm. The designed concrete strength grade is C30, with a slab thickness of 150 mm.

During on-site testing, a total of six measuring zones were set up for concrete strength rebound, with 16 measuring points in each zone. Two measuring points were used for concrete carbonation depth testing, and each measuring point was arranged in a cross shape. There are 3 measurement areas for the thickness of the concrete protective layer, each containing 6 measurement points. The crack width was measured at three measuring points in the middle of the component. After on-site testing, the average rebound strength of concrete is 44.0 MPa, and the average carbonation depth is 0.15 mm. According to the Technical Specification for Inspecting of Concrete Compressive Strength by Rebound Method [36], the converted concrete strength is 50.20 Mpa. The average thickness of the concrete protective layer is 25.9 mm. The average crack width is 0.17 mm. The diameters of corroded steel bars at three measuring points are 13.09 mm, 13.19 mm, and 13.44 mm, respectively. Figs 15–18 show the on-site inspection results.

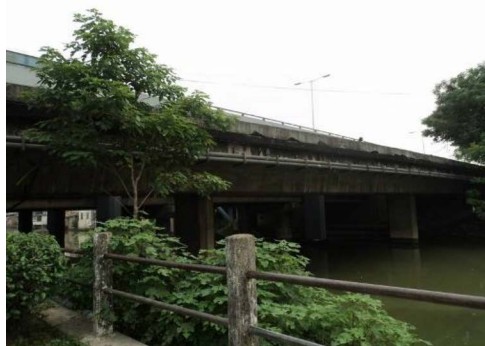

**Fig 14. Scene of the Sanhongqi Overpass.**

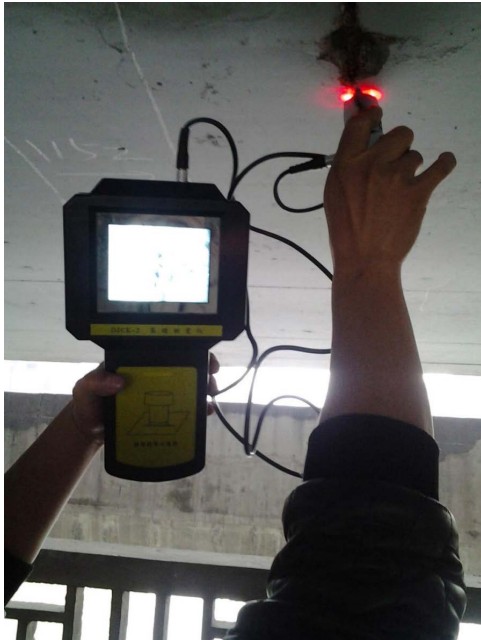

**Fig 15. Crack width detection.**

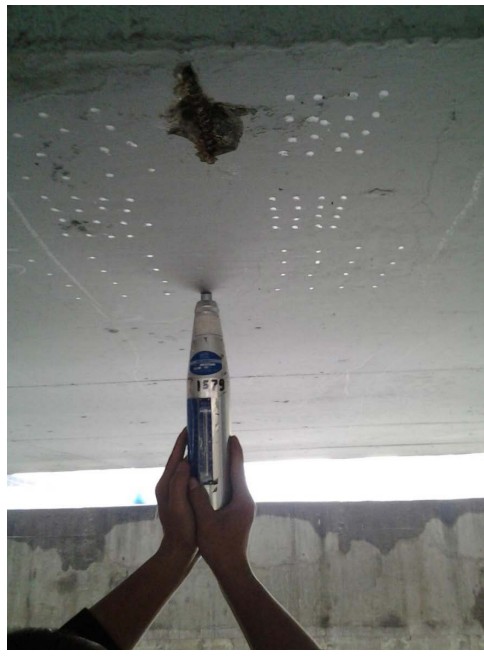

**Fig 16. Concrete strength test.**

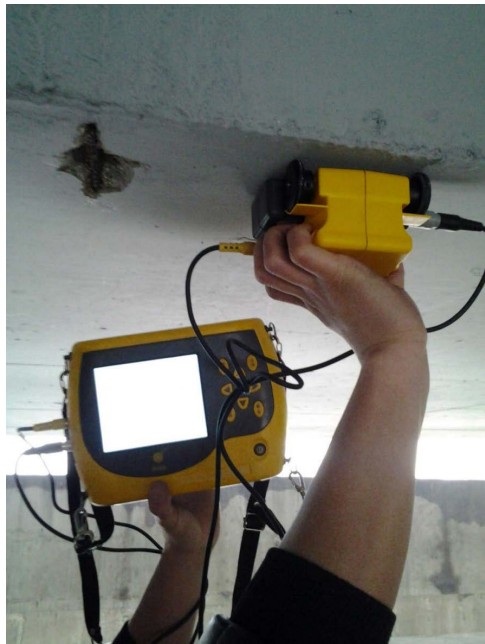

**Fig 17. Testing of concrete protective layer.**

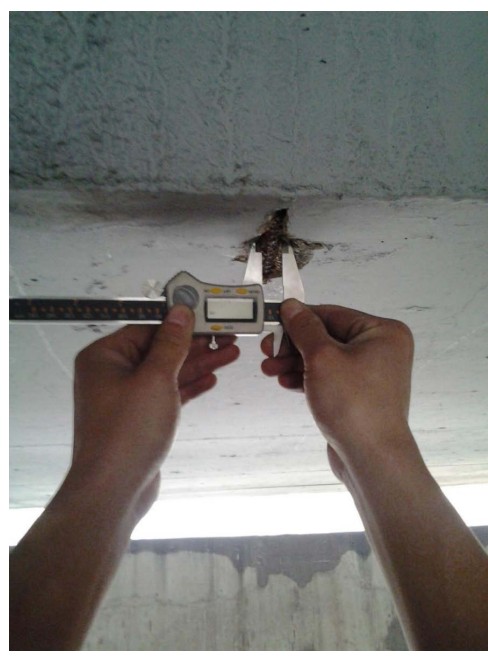

**Fig 18. Diameter test of corroded steel bars.**

The model was applied to the Sanhongqi Overpass, where the theoretical corrosion rate was calculated as 9.20%, compared to the measured value of 10.56%, with a deviation of 12.88%. This result confirms the model's applicability to real-world scenarios.

To improve accuracy in practical applications, a correction factor between 1.05 and 1.15 is recommended, based on the specific design parameters of the structure, which can meet the needs of rapid non-destructive testing technology in practical engineering.While the model shows high accuracy in predicting corrosion rates, factors such as environmental variability and external stresses were not considered and should be addressed in future studies.

## 5 Conclusion

This study presents a comprehensive analysis of the relationship between surface crack width and reinforcement corrosion in reinforced concrete slabs. By incorporating the influence of distributed reinforcement and protective layer thickness, the proposed model achieves high predictive accuracy, as confirmed by experimental and practical validation. The proposed model offers a valuable tool for engineers to assess and mitigate corrosion in reinforced concrete structures, supporting proactive maintenance and extending service life. In addition, the corresponding semi-theoretical and semi-empirical model is given based on the existing research results, as well as the theoretical calculated values are compared with the experimental values. The key contributions of this work include 1) Introducing a modified stiffness reduction factor for improved prediction accuracy, 2) Demonstrating the significant role of distributed reinforcement in reducing corrosion rates and crack propagation, and 3) Providing practical recommendations for optimizing structural durability in chloride environments. The conclusions are as follows:

For the concrete slab without distributed reinforcement, the reinforcement in the specimens can be regarded as uniformly corroded with the accelerated rusting test by electrification. After cracking of the direction, the cracking width along the reinforcement is relatively uniform in the whole length direction, and the width of the cracks along the reinforcement basically does not differ much. As the thickness of the protective layer is larger, the corresponding amount of reinforcement corrosion and the width of surface cracks are smaller. For the concrete slab with distributed reinforcement, due to the inconsistent spacing of distributed reinforcement, after cracking of the specimen, the cracking width along the reinforcement is not uniform in the length direction, and as the spacing of distributed reinforcement is larger, the corresponding amount of reinforcement corrosion and the width of surface cracks is also larger.The theoretical formula given in this paper is small than the actual detection value, in order to make the prediction results more accurate and improve the detection efficiency, the calculation model of this paper can be corrected and the correction coefficient value range can be taken as 1.05～1.15.

While the model demonstrates high accuracy, factors such as environmental variability and the effects of external loads were not addressed and should be explored in future research. Future studies could focus on integrating additional variables, such as long-term environmental effects and multi-directional stress states, to further enhance the model's applicability. These findings contribute to advancing non-destructive testing techniques and improving the design and maintenance of reinforced concrete structures, addressing critical challenges in modern civil engineering.

## Supporting information

**S1 Table. Effects of protective layer thickness (I# slab).**
(DOCX)

**S2 Table. Influence of spacing between distributed reinforcement (II# slab).**
(DOCX)

**S3 Table. Analysis of the model calculated values and test data values of I# slab and II# slab.**
(DOCX)

**S1 Figure. Crack width test on board surface.**
(DOCX)

## Author contributions

**Conceptualization:** Shangchuan Zhao.

**Data curation:** Duo Wu, Hao Wu.

**Formal analysis:** Weihong Wan.

**Funding acquisition:** Jian Cao.

**Investigation:** Weihong Wan.

**Methodology:** Duo Wu.

**Project administration:** Jian Cao.

**Resources:** Shangchuan Zhao.

**Software:** Hao Wu.

**Supervision:** Jian Cao.

**Writing – original draft:** Ziyi Zou.

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
