## [Decision Letter · Decision Letter 0]

29 Jan 2025

PONE-D-25-02651Calculation of Steel Corrosion Rate of Reinforced Concrete Slab Based on Rust Expansion CrackPLOS ONE

Dear Dr. Cao,

Thank you for submitting your manuscript to PLOS ONE. After careful consideration, we feel that it has merit but does not fully meet PLOS ONE’s publication criteria as it currently stands. Therefore, we invite you to submit a revised version of the manuscript that addresses the points raised during the review process.

**ACADEMIC EDITOR: ** Dear Authors;

   Based on the Reviewers’ comments, it is hardly suggested that the proposed

Manuscript be meticulously revised and improved. It is suggested that all points indicated by Reviewers be individually solved and improved adequately.==============================

We look forward to receiving your revised manuscript.

Kind regards,

Wislei Riuper Osório

Academic Editor

PLOS ONE

Additional Editor Comments :

Dear Authors;

Based on the Reviewers’ comments, it is hardly suggested that the proposed

Manuscript be meticulously revised and improved. It is suggested that all points indicated by Reviewers be individually solved and improved adequately.

Reviewers' comments:

Reviewer's Responses to Questions

**Comments to the Author**

1. Is the manuscript technically sound, and do the data support the conclusions?

Reviewer #1: Partly

Reviewer #2: Partly

2. Has the statistical analysis been performed appropriately and rigorously? 

Reviewer #1: No

Reviewer #2: I Don't Know

3. Have the authors made all data underlying the findings in their manuscript fully available?

Reviewer #1: Yes

Reviewer #2: Yes

4. Is the manuscript presented in an intelligible fashion and written in standard English?

Reviewer #1: Yes

Reviewer #2: No

5. Review Comments to the Author

Reviewer #1: The attached document contains a detailed review of the manuscript, including specific suggestions and comments for improvement across all sections: Introduction, Modeling, Experiment, Results of the Experiment, Modeling Verification, and Conclusion.

The review emphasizes the need for:

A stronger contextualization and clearer problem statement in the introduction.

Improved clarity and emphasis on the novelty of the proposed model in the modeling section.

Detailed procedural descriptions and connections to practical engineering in the experiment section.

A structured presentation of findings and implications in the results and conclusion sections.

Please refer to the attached document for a comprehensive review and detailed recommendations for each section of the manuscript.

Reviewer #2: The authors present an interesting approach to the study of corrosion in slabs. However, some points need to be improved. The literature review should be more up-to-date. Papers published after 2023 are recommended. The introduction should provide information that is more focused on the problem that the authors want to answer. The figures used in the paper are not of good quality and do not speak for themselves. Tables and figures should contain information about the statistical analysis, such as error bars. It is recommended that the conclusion be written in a continuous text, not in items. After a major revision, the paper should undergo a new peer review.

6. PLOS authors have the option to publish the peer review history of their article (what does this mean? ). If published, this will include your full peer review and any attached files.

**Do you want your identity to be public for this peer review?** For information about this choice, including consent withdrawal, please see our Privacy Policy .

Reviewer #1: **Yes: ** Prof. Dr. Yuri Alexandre Meyer

Reviewer #2: No

---

## [Author Response · Author response to Decision Letter 1]

18 Mar 2025

Thank you for reviewing our manuscript and for your valuable comments. We take these suggestions very seriously and have revised the manuscript accordingly based on your feedback. The attached file named "Respond to Reviewers" is our itemized response to the review comments.

---

## [Editor Report · Decision Letter 1]

19 Mar 2025

Calculation of Steel Corrosion Rate of Reinforced Concrete Slab Based on Rust Expansion Crack

PONE-D-25-02651R1

Dear Dr. Cao,

We’re pleased to inform you that your manuscript has been judged scientifically suitable for publication and will be formally accepted for publication once it meets all outstanding technical requirements.

Kind regards,

Wislei Riuper Osório

Academic Editor

PLOS ONE

Additional Editor Comments (optional):

Based on the all rebuttals and comments provided, it is observed that the revised version of the manuscript deserves its final publication. In this sense, I am indicating this manuscript to its final publication. It is only and finally suggested that a revision in Grammar and Spelling be carried out into the proof versioN.
---

## [Editor Report · Acceptance letter]

PONE-D-25-02651R1

PLOS ONE

Dear Dr. Cao,

I'm pleased to inform you that your manuscript has been deemed suitable for publication in PLOS ONE. Congratulations! Your manuscript is now being handed over to our production team.

Kind regards,

on behalf of

Dr. Wislei Riuper Osório

Academic Editor

PLOS ONE